# Main constraints for RNAi induced by expressed long dsRNA in mouse cells

Tomas Demeter[1],[*], Michaela Vaskovicova[1],[*], Radek Malik[1], Filip Horvat[1,2], Josef Pasulka[1], Eliska Svobodova[1], Matyas Flemr[1], Petr Svoboda[1]

RNAi is the sequence-specific mRNA degradation guided by siRNAs produced from long dsRNA by RNase Dicer. Proteins executing RNAi are present in mammalian cells but rather sustain the microRNA pathway. Aiming for a systematic analysis of mammalian RNAi, we report here that the main bottleneck for RNAi efficiency is the production of functional siRNAs, which integrates Dicer activity, dsRNA structure, and siRNA targeting efficiency. Unexpectedly, increased expression of Dicer cofactors TARBP2 or PACT reduces RNAi but not microRNA function. Elimination of protein kinase R, a key dsRNA sensor in the interferon response, had minimal positive effects on RNAi activity in fibroblasts. Without high Dicer activity, RNAi can still occur when the initial Dicer cleavage of the substrate yields an efficient siRNA. Efficient mammalian RNAi may use substrates with some features of microRNA precursors, merging both pathways even more than previously suggested. Although optimized endogenous Dicer substrates mimicking miRNA features could evolve for endogenous regulations, the same principles would make antiviral RNAi inefficient as viruses would adapt to avoid efficacy.

## Introduction

dsRNA, a helical structure formed by complementary antiparallel RNA strands, has important biological effects. dsRNA can arise via (1) base-pairing of complementary sequences in RNA molecule(s) or (2) second strand synthesis by an RNA-dependent RNA polymerase (RdRP). Although mammals lack endogenous RdRPs (Stein et al, 2003a), dsRNA can still be produced by viral RdRPs in infected cells. In mammalian cells, dsRNA can undergo conversion of adenosines to inosines by adenosine deaminases acting on RNA (ADAR), induce a sequence-independent IFN response, or induce sequence-specific RNAi.

The interferon response is a complex innate immunity system where multiple sensors converge on a response involving activation of NFκB transcription factor and interferon-stimulated genes (Geiss et al, 2001). The key dsRNA sensor in the IFN response is protein kinase R (PKR, reviewed in Sadler & Williams, 2007), which is activated by dsRNA and inhibits translation initiation through phosphorylation of the α-subunit of eukaryotic initiation factor 2 (eIF2α) (Farrell et al, 1978; Meurs et al, 1990). PKR response is sequence independent and affects translation universally although inhibition restricted to specific mRNAs was also observed (Kaufman et al, 1989; Ben-Asouli et al, 2002; Nejepinska et al, 2014). In addition to PKR, other factors sensing dsRNA contribute to the IFN response, such as RIG-I–like receptors (RIG-I, MDA5, and LGP2, reviewed in Lassig & Hopfner, 2017) or oligoadenylate synthetases, which yield 2′,5′-oligoadenylate triggers for global RNA destabilization by RNase L (reviewed in Kristiansen et al, 2011).

RNAi has been defined as sequence-specific RNA degradation induced by long dsRNA (Fire et al, 1998). During canonical RNAi, long dsRNA is cut by RNase III Dicer into ~22 nt siRNAs, which are bound by an Argonaute (AGO) endonuclease, thus forming an RNA-induced silencing complex (RISC). siRNAs guide sequence-specific mRNA recognition and endonucleolytic cleavage in the middle of base-paring between siRNA and mRNA molecules (reviewed in Nejepinska et al, 2012a). Additional factors participating in RNAi include dsRNA-binding proteins (dsRBP). In *Drosophila*, R2D2 dsRBP restricts Dicer specificity to long dsRNA (Cenik et al, 2011; Nishida et al, 2013; Fukunaga & Zamore, 2014). Mammalian dsRBPs TARBP2 and PACT, which interact with Dicer during small RNA loading (Chendrimada et al, 2005; Haase et al, 2005), have unknown roles in routing long dsRNA into RNAi and IFN pathways in vivo.

Mammalian genomes encode proteins necessary and sufficient for reconstituting canonical RNAi in vitro (MacRae et al, 2008) or in yeast (Suk et al, 2011; Wang et al, 2013). However, mammalian proteins primarily function in a gene-regulating microRNA pathway (reviewed in Bartel, 2018), whereas only negligible amounts of siRNAs of unclear functional significance are typically observed in mammalian cells (reviewed in Svoboda, 2014). At the same time,

[1]Institute of Molecular Genetics, Academy of Sciences of the Czech Republic, Prague, Czech Republic  [2]Bioinformatics Group, Division of Molecular Biology, Department of Biology, Faculty of Science, University of Zagreb, Zagreb, Croatia

Correspondence: svobodap@img.cas.cz
*Tomas Demeter and Michaela Vaskovicova contributed equally to this work.

successful RNAi has been occasionally experimentally achieved in cultured cells with different types of long dsRNA molecules, including transfection of long dsRNA into embryonic stem cells (ESC) and embryonic carcinoma cells (Billy et al, 2001; Yang et al, 2001; Paddison et al, 2002), and expression of various types of long dsRNA in ESCs, transformed, and primary somatic cells (Gan et al, 2002; Paddison et al, 2002; Diallo et al, 2003; Shinagawa & Ishii, 2003; Wang et al, 2003; Yi et al, 2003; Gantier et al, 2007).

One of the factors limiting mammalian RNAi is the mammalian Dicer, which does not produce siRNAs from long dsRNA substrates efficiently (reviewed in Svobodova et al, 2016). One of the ways to achieve high siRNA production is truncation of Dicer at its N terminus, which increases siRNA generation in vitro (Ma et al, 2008) and in cultured cells (Kennedy et al, 2015). An N terminally truncated Dicer variant occurs naturally in mouse oocytes (Flemr et al, 2013), the only known mammalian cell type where RNAi is highly active and functionally important. Another factor impeding RNAi is the presence of the IFN pathway as evidenced by enhanced siRNA production and RNAi activity upon inactivation of different components of the IFN pathway, such as dsRNA sensor PKR, RIG-I–like receptor LGP2 (Kennedy et al, 2015; van der Veen et al, 2018), or mediators MAVS or IFNAR1 (Maillard et al, 2016).

The mammalian RNAi has a known biological role in mouse oocytes where it suppresses mobile elements and regulates gene expression (Murchison et al, 2007; Tang et al, 2007; Tam et al, 2008; Watanabe et al, 2008). However, there is scarce and unclear evidence for biological significance of mammalian RNAi elsewhere. A part of Dicer loss-of-function phenotype in murine ESCs has been attributed to the lack of endogenous RNAi (Babiarz et al, 2008). However, reported ESC endo-siRNAs, such as those derived from a hairpin-forming B1/Alu subclass of short interspersed elements (SINEs), resemble non-canonical microRNAs rather than a siRNA population produced from a long dsRNA (Flemr et al, 2013). Similarly, hippocampal endo-siRNAs emerging from the SyngaP1 locus (Smalheiser et al, 2011) map to a sequence that is nowadays annotated as a microRNA locus (Kozomara & Griffiths-Jones, 2014).

It has been questioned whether or not could endogenous RNAi contribute to mammalian antiviral defense (reviewed in Cullen et al, 2013; Gantier, 2014). In contrast to invertebrates, data supporting direct involvement of mammalian RNAi in antiviral defense are rather inconclusive. Although several studies suggested that RNAi could provide an effective antiviral response in ESCs and in mouse embryos (Li et al, 2013, 2016; Maillard et al, 2013; Qiu et al, 2017), other studies did not support that notion. For example, no siRNAs of viral origin have been found in human cells infected with a wide range of viruses (Pfeffer et al, 2005) or present siRNAs were not sufficient to mediate effective RNAi (Tsai et al, 2018). Although it seems unlikely that mammalian RNAi would be a substantial antiviral mechanism co-existing with long dsRNA-induced IFN response, conditions still remain unclear, under which mammalian RNAi could operate effectively.

To bring more systematic insights into disparities concerning observable RNAi activity in mammalian cells, we analyzed RNAi induction in mouse fibroblasts and ESCs using a set of plasmids expressing different types of dsRNA, which could target luciferase reporters with complementary sequences. We show that endogenous RNAi in mouse cells is severely restricted at multiple levels. Mouse fibroblasts and ESCs contain minimal amount of endogenous dsRNA that could be converted into endo-siRNAs and, at the same time,

inefficiently convert expressed dsRNA to siRNAs. This owes to low Dicer activity, which is able to generate only limited amounts of siRNA from dsRNA. Although increased Dicer activity stimulates RNAi, increased expression of dsRBPs TARBP2 or PACT reduces RNAi without affecting the miRNA pathway. However, if the substrate contains a blunt terminus, RNAi can be effective if the first cleavage by Dicer (yields an effective siRNA. This implies that RNA plays a secondary role, if any, in antiviral response because viruses could, among other adaptations, escape RNAi through evolving terminal sequences upon selection against providing a source of efficient siRNAs.

## Results and Discussion

To investigate mammalian RNAi, we expanded a long RNA hairpin expression system originally developed for transgenic RNAi in mice (reviewed in Malik & Svoboda, 2012). It combines (i) an inverted repeat producing a long (>400 bp) dsRNA hairpin inserted into the 3′UTR of an EGFP reporter and (ii) *Renilla* (RL) and firefly luciferase (FL) reporters for distinguishing sequence-specific and sequence-independent effects (Fig 1A). The hairpin plasmids were derived from *Mos*, *Elavl2*, and *Lin28a/b* mRNA sequences (Fig S1A) and, for brevity, are referred to as MosIR, Lin28IR, and Elavl2IR. The long hairpin RNA organization is similar to some naturally occurring long dsRNA hairpins, which give rise to endogenous siRNAs in *Caenorhabditis elegans* (Morse & Bass, 1999) and mouse oocytes (Tam et al, 2008; Watanabe et al, 2008). Importantly, all three hairpin transcripts could be efficiently immunoprecipitated with an anti-dsRNA antibody (Nejepinska et al, 2014) and their expression induced robust RNAi in oocytes in vivo (Stein et al, 2003b; Chalupnikova et al, 2014; Flemr et al, 2014). In a control plasmid CAG-EGFP-MosMos (Fig 1A, referred to as MosMos hereafter), the *Mos* tandem sequence is oriented head-to-tail; hence, the plasmid has the same size and nucleotide composition as MosIR but does not produce dsRNA. Targeted RL reporters were derived from a *Renilla* luciferase expression plasmid by inserting *Mos*, *Lin28*, or *Elavl2* sequences in the 3′UTR. A common FL reporter serves as a nontargeted control (in sequence-specific context). dsRNA expression and RNAi activity were analyzed in mouse ESCs and NIH 3T3 (referred to as 3T3 hereafter) mouse fibroblasts (Todaro & Green, 1963), which represent undifferentiated and differentiated cell types, respectively.

In a typical experiment, a dsRNA-expressing plasmid (e.g., MosIR) and two luciferase reporters (and, eventually, another tested factor) were transiently co-transfected, and luciferase activities were quantified 48 h later. Sequence-specific and sequence-independent effects could be distinguished in samples transfected with MosMos (negative control), MosIR (targeting dsRNA), or Elavl2IR (nontargeting dsRNA—a positive control for nonspecific dsRNA effects) by comparing RL-Mos (MosIR-targeted) and FL (nontargeted) reporter activities. Importantly, sequence-independent dsRNA effects, which strongly reduce raw activities of both luciferase reporters in transfected cells (Fig 1B), are not apparent in normalized data, which are typically displayed as a targeted reporter signal divided by the nontargeted reporter signal (here RL-Mos/FL, Figs 1C and S1B). We have shown previously that reporter expression from co-transfected plasmids is particularly inhibited in a PKR-dependent

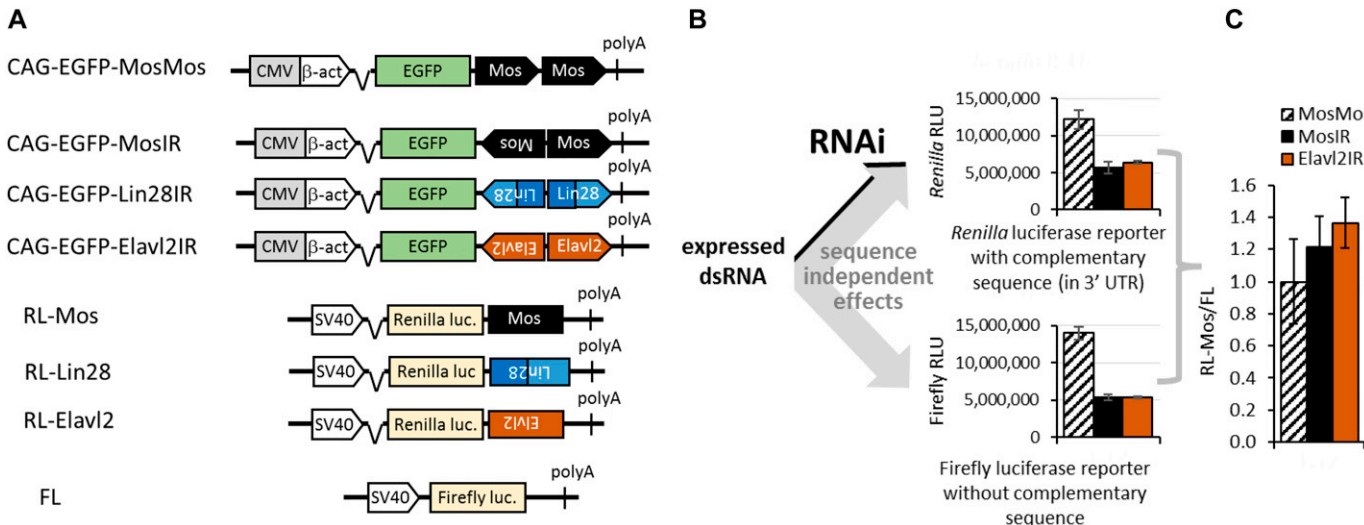

**Figure 1. Long dsRNA expression system for cultured mammalian cells.**
**(A)** Schematic depiction of relevant parts of plasmids used for dsRNA expression and detection of sequence-specific and sequence-independent effects. **(B, C)** Scheme of experimental design with a typical result obtained in mammalian somatic cells. 3T3 cells were transfected with MosIR and luciferase reporters, and luciferase activities were analyzed 48 h after transfection. Note that normalization of RL-Mos reporter activity using a nontargeted firefly luciferase (FL) reporter (RL-Mos/FL graph) in the panel (C) masks sequence-independent effects, which are apparent in raw luciferase data before normalization. LacZ-expressing plasmid was co-transfected as a "neutral" co-expressed protein, which was later used as a control. LacZ yielded the same sequence-independent dsRNA effect as previously observed without a tested co-expressed protein (Nejepinska et al, 2014) and similar effect as expression of EGFP and RFP (Fig S1B). Firefly and *Renilla* graphs depict raw values from one transfection experiment performed in triplicates, the graph in the panel (C) data from four independent experiments performed in triplicates. Error bars = standard deviation (SD).

manner by dsRNA expressed from a co-transfected plasmid (Nejepinska et al, 2014). The occurrence of this effect in raw data is, thus, a good indicator of dsRNA expression.

The absence of RNAi effect in 3T3 cells (Figs 1C and S1B) could be expected as RNAi was absent in somatic cells of mice ubiquitously expressing MosIR (Nejepinska et al, 2012b) despite MosIR inducing highly specific RNAi effect in mouse oocytes (Stein et al, 2003b, 2005; Nejepinska et al, 2012b). At the same time, expression of a different type of long dsRNA in 3T3 cells was reported to induce RNAi (Wang et al, 2003), suggesting that conditions exists, under which RNAi could operate in 3T3 cells. Thus, MosIR expression in 3T3 cell culture seemed to be a good starting model for exploring constraints for functional mammalian RNAi.

### Inefficient siRNA production from long dsRNA in mouse cells

To examine the cause of inefficient RNAi, we first examined *Mos* siRNA levels in 3T3 cells transfected with MosIR using small RNA sequencing (RNA-seq). We also co-transfected plasmids expressing either full-length Dicer expressed in somatic cells (denoted Dicer[S]) or the truncated Dicer isoform supporting RNAi in mouse oocytes (denoted Dicer[O]). The experiment yielded reproducible small RNA populations comparable with results of an earlier RNA-seq analysis of ESCs (Flemr et al, 2013) (Fig S2A). MosIR expression in normal 3T3 cells yielded only minimal amounts of 21–23 nt siRNAs (Fig 2A, ~100 reads per million [RPM] when normalizing the read abundance to the entire small RNA library). Relative to 3T3 cells transfected with MosIR, co-expression of Dicer[S] or Dicer[O] increased Mos 21–23 nt siRNA production 5.7× or 24.2×, respectively (Fig 2A). siRNA levels in libraries from Dicer[O]-expressing 3T3 cells were similar to the earlier analysis of Dicer[O]-expressing ESCs (Fig 2B).

Normal 3T3 cells, thus, have a minimal capability to produce siRNAs from long dsRNA, which has termini formed of longer single-stranded RNAs or a loop, hence inaccessible for Dicer. This is consistent with the previous observation that human Dicer efficiently cleaves dsRNA with blunt ends or 2-nt 3'-overhangs from its termini and less efficiently inside the duplex (Zhang et al, 2002). siRNA production could be improved by increasing levels of full-length Dicer (Fig 2A). Remarkably, relative siRNA distribution along the *Mos* sequence was almost identical in 3T3 cells expressing additional Dicer[S] and cells expressing Dicer[O] (Fig S2B). The pattern was not specific to 3T3 cells because it was observed also in ESCs expressing Dicer[O] (Fig S2B). This implies that cells expressing high levels of full-length Dicer could also generate more siRNAs from long dsRNA. Dicer expression varies across cell types. Off note is that the highest Dicer mRNA levels were found in oocytes, lymphocytes, and mast cells (Wu et al, 2016).

Another notable observation was that normal 3T3 cells essentially lacked endogenous siRNAs (Fig 2C). We inspected loci in 3T3 cells giving rise to 21–23-nt sequences using the same algorithm as in our previous study in ESCs (Fig 2D), which revealed a small number of loci giving rise to long dsRNA, which was converted to siRNAs by Dicer[O] (Flemr et al, 2013). However, we did not find any locus in 3T3 cells that would produce an apparent population of endo-siRNAs from long dsRNA like in ESCs. Instead, all genomic loci with higher abundance of perfectly mapped 21–23-nt RNAs in Dicer[O]-expressing cells were reminiscent of miRNA loci (e.g., Fig S2C). This contrasts with ESCs expressing Dicer[O], where a distinct population of loci generating siRNA pools >100 RPM was observed (Fig 2D).

We examined small RNAs derived from repetitive mobile elements separately, grouping all 21–23-nt reads according to mapping

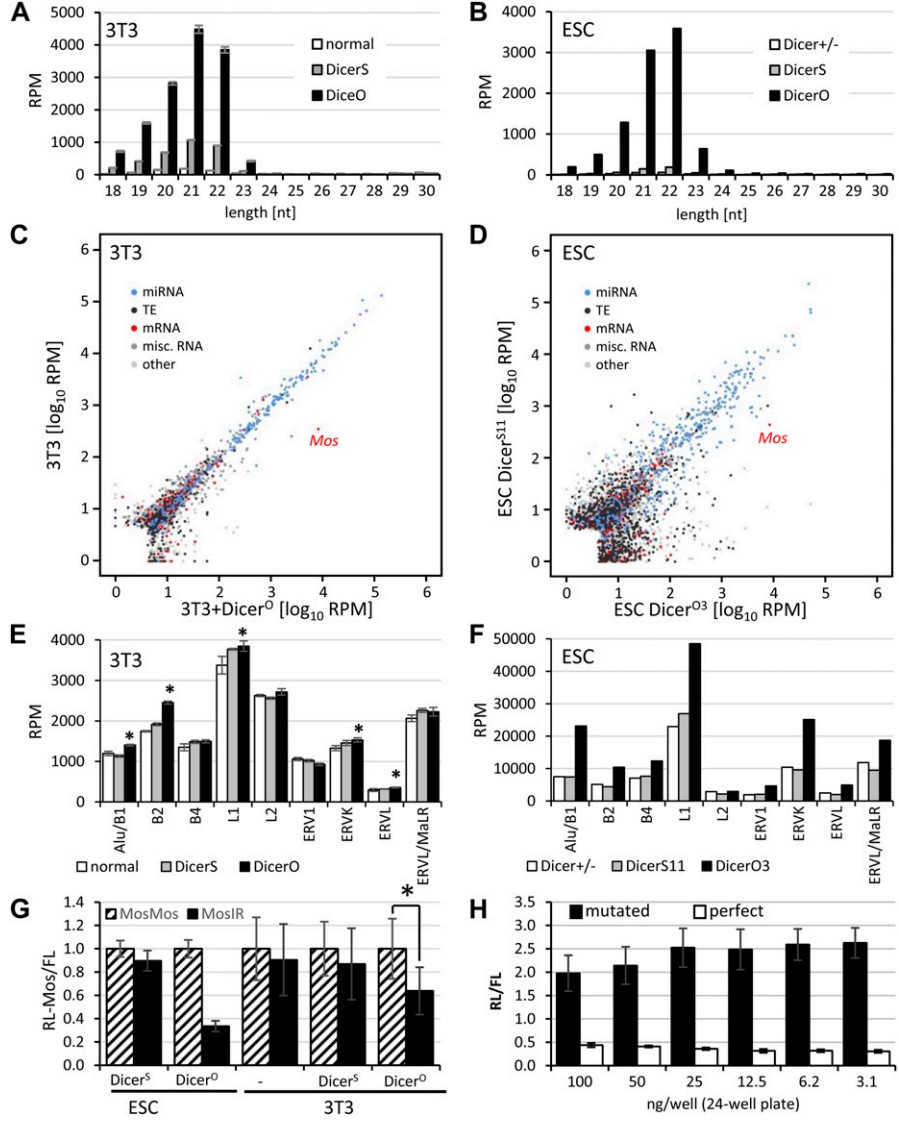

**Figure 2.   21–23-nt RNAs in cells expressing Mos dsRNA and Dicer variants.**
**(A)** Mos siRNA abundance in 3T3 cells reflects Dicer activity. 3T3 cells were transfected either with a plasmid expressing Dicer[S] or Dicer[O] and small RNA content was analyzed by RNA-seq. Shown is abundance of small RNAs of different lengths perfectly matching to the genome expressed as RPM of mapped reads (perfect match) from each RNA-seq library. Libraries were prepared in triplicates from separately transfected wells. Error bars = SD. **(B)** Mos siRNA abundance in different ESC lines. RNA-seq data from Flemr et al (2013) were mapped onto the mm10 mouse genome version. The values are averages of biological duplicates (lines O3 and O4 for Dicer[O] and S2 and S11 for Dicer[S]). **(C, D)** Quantitative display of small RNA clusters in 3T3 cells (panel C) and ESCs (panel D) expressing full-length (Dicer[S]) and truncated Dicer variant (Dicer[O]). For 3T3 cells, data depict comparison of 21–23-nt RNA-producing loci in normal 3T3 cells and 3T3 cells transfected with a Dicer[O]-expressing plasmid. Both samples were also transfected with MosIR. Mapping small RNAs to the genome and identification and annotation of clusters of 21–23-nt small RNAs was performed as described previously (Flemr et al, 2013). Each cluster is represented as a colored point indicating its category. Position of a cluster corresponds to the median log10 of RPM value in the analyzed samples. ESC data were taken from Dicer[O-3] and Dicer[S-11] RNA-seq from our previous work (Flemr et al, 2013) and were remapped to the current version of the mouse genome and classified into specific categories. Specifically indicated are positions of MosIR siRNA pools. **(E)** 21–23-nt RNAs derived from different classes of mobile elements in 3T3 cells. Error bars = SD. Although the relative changes were minimal, in some cases putative endo-siRNA levels were significantly higher in Dicer[O]-expressing cells ($P$-value < 0.05, indicated by asterisks). **(F)** 21–23-nt RNAs derived from different classes of mobile elements in ESC lines. Shown is an average of duplicate libraries used for constructing the panel (D). **(G)** Expression of Dicer[O] is sufficient for inducing RNAi effect on RL-Mos reporter in MosIR-transfected ESCs and 3T3 cells. RL-Mos reporter activity in MosIR-expressing cells is shown relative to RL-Mos reporter activity in cells co-transfected with MosMos-expressing plasmid, which was set to one. The experiment was performed twice in triplicate transfections. Error bars = SD. **(H)** Efficient repression of *Renilla* luciferase reporter carrying a single perfectly complementary site for miR-30C. A plasmid carrying three mutated miR-30–binding sites was used as a control. Indicated amounts of *Renilla* and firefly reporters were transfected into 3T3 cells. Results are present as a simple ratio of *Renilla* luciferase activity divided by nontargeted firefly luciferase activity. The transfection experiment was performed in a triplicate. Error bars = SD.

to specific retrotransposon groups (Fig 2E). This analysis showed in several cases (SINE B2, LINE1, ERVK, and ERVL) a minor increase in Dicer[O]-expressing cells (10–20%, up to several hundred RPM difference). However, it is unclear, what type of RNA substrates was responsible for this increase. We showed that transcribed inverted repeats of Alu/SINE B1 can form substrates producing small RNAs appearing more as non-canonical miRNAs than siRNA pools (Flemr et al, 2013). Furthermore, 21–23-nt populations mapping to the same elements in ESCs show almost an order of magnitude higher abundance and much stronger increase in Dicer[O]-expressing cells (Fig 2F), which likely stems from open chromatin structure and dsRNA production in ESCs (Martens et al, 2005).

Altogether, these data show that 3T3 cells neither produce significant amounts of dsRNA nor possess robust Dicer activity that could process it. Regardless whether dsRNA absence in 3T3 cells is due to minimal dsRNA production or its efficient removal by other pathways, the endogenous RNAi is apparently not operating in 3T3 cells. RNAi can be revived in 3T3 cells through strongly increasing Dicer activity by expressing Dicer[O] and providing a dsRNA substrate (Fig 2G). However, in contrast to ESCs expressing Dicer[O], weaker (36%) sequence-specific repression of RL-Mos was observed despite *Mos* siRNA levels in 3T3 cells transfected with MosIR and Dicer[O]-expressing plasmid were comparable with those observed in ESCs stably expressing Dicer[O] (Fig 2A and B).

Notably, combined ~8,000 RPM abundance of 21- and 22-nt *Mos* siRNAs in Dicer[O]-transfected 3T3 cells (Fig 2A) would reach RPM values equivalent to highly abundant miRNAs. Although RPM values from RNA-seq data are not a reliable predictor of miRNA abundance because RNA-seq protocols may introduce biases (Linsen et al, 2009), high RPM values generally indicate higher miRNA abundance.

Among the most abundant miRNAs in 3T3 cells was the *Let-7* family, abundance of all *Let-7* miRNAs added up to 11,900 RPM, the most abundant member *Let-7f* was at ~4,300 RPMs, and Let-7a reached ~2,100 RPM. Similarly, abundancies of all *miR-30* family members added up to 6,100 RPM, and the most abundant member *miR-30c* was at ~2,100 RPM (corresponding to 0.7% of the first 30 most abundant miRNAs).

To examine small RNA-mediated cleavage of cognate RNAs in 3T3 cells, we analyzed the ability of miR-30 to repress a target with a single perfectly complementary binding site. We previously produced luciferase reporters with a single miRNA binding site with perfectl complementarity to miR-30c (miR-30 1xP) for monitoring RNAi-like cleavage by endogenous miRNAs (Ma et al, 2010). Earlier reporter testing included 3T3 cells, but those experiments were performed with very low amounts of transfected reporters (1 ng/well in a 24-well plate). Thus, it was unclear whether the reporter would be efficiently repressed when higher amounts would be transfected. Accordingly, we titrated the miR-30 1xP reporter up to 100 ng/well in a 24-well plate. As a control, we used a reporter with three mutated miR-30c binding sites (miR-30 3xM), which should not be repressed by endogenous miRNAs. Although we could observe less efficient repression with increasing amount of miR-30 reporters, a single perfect miR-30c binding site was still sufficient to lower the miR-30 1xP reporter to 22% of miR-30 3xM at 100 ng/well (Fig 2H).

It was rather unexpected that ~8,000 RPM of 21–22-nt siRNAs induced only mild repression of the reporter (Fig 2G). The supply of AGO proteins is probably not the main limiting factor as endogenous AGO-loaded miR-30 is sufficient for repression of miR-30 1xP reporter. Inefficient RISC loading of MosIR siRNAs is also unlikely because of highly similar profiles of 21–22-nt small RNAs mapped onto the MosIR sequence (Fig S2B). Inefficient RISC loading would be accompanied by accumulation of siRNA duplexes. However, RNA-seq data suggest that passenger strands of siRNAs do not seem to accumulate in the samples we sequenced.

Importantly, an apparent high abundance of MosIR siRNAs in 3T3 cells is overestimated for two reasons. First, just a half of the loaded MosIR siRNAs would be antisense siRNAs able to cleave the RL-Mos reporter. Second, only a fraction of complementary siRNAs may be engaged in effective RL-Mos repression because, for example, secondary RNA structures prevent efficient targeting (Ameres et al, 2007). In addition, 3T3 cells have higher transfection efficiency than ESCs. Thus, similar RPM levels in RNA-seq libraries from transiently transfected 3T3 cells and ESCs would mean that ESCs have higher siRNA levels per cell. Some bias might also come from heterogeneity in transiently transfected cell population. Finally, we cannot rule out that poorly folded MosIR transcripts could serve as a decoy and reduce efficiency of targeting. At the same time, the Mos-based RNAi system is excellent for studying constraints for endogenous RNAi because it requires highly optimized RNAi activity, whereas Lin28-based system (Fig 4C) appears to be more sensitive for detecting RNAi effects.

In any case, the threshold for efficient reporter repression by MosIR-derived siRNA population appears relatively high in cultured cells. MosIR did not induce significant RNAi in normal 3T3 cells nor in ESCs (~10% knockdown). Introduction of Dicer$^O$, a truncated Dicer

variant supporting RNAi in mouse oocytes, was sufficient to enhance siRNA production and, in the case of ESCs, also to induce RNAi effect (~65% knockdown of RL-Mos reporter activity). At the same time, our data imply that cells expressing full-length Dicer may not be able to mount efficient RNAi in the presence of an excess of dsRNA substrate, which does not have Dicer-accessible termini. Importantly, we also observed sequence-independent dsRNA effects in ESCs transfected with MosIR despite ESCs are reported to lack the IFN response (D'Angelo et al, 2017).

## Effects of expression dsRBPs on RNAi

To further investigate constrains for RNAi in mouse cells, we examined effects of different dsRBPs on RNAi. These included PKR, which was used as a positive control expected to interfere with RNAi and Dicer-binding partners TARBP2 and PACT (Chendrimada et al, 2005; Haase et al, 2005; Laraki et al, 2008), which could potentially support RNAi. We did not include analysis of adenosine deamination. First, although ADAR1 was previously shown to edit endogenous dsRNA to prevent its sensing by MDA5 (Liddicoat et al, 2015), sequences of small RNAs from MosIR did not suggest editing would be a significant constraint in ESCs and 3T3 cells. Second, ADAR1 overexpression would be unlikely to positively affect RNAi (Reich et al, 2018). As a negative control, we used expression of LacZ.

We did not observe any positive effect of any dsRBP on RNAi (Fig 3A), whereas immunoprecipitation showed approximately 50-fold enrichment of TARBP2 and PACT binding to MosIR RNA compared with a control protein (Fig S3). Remarkably, expression of TARBP or PACT counteracted sequence-independent effects of dsRNA expression, which indicates that both expressed proteins are binding MosIR hairpin and compete with endogenous PKR binding (Fig 3A, lower part). Consistent with this notion, ectopic PKR expression further increased sequence-independent repression of luciferase reporters, including MosMos-transfected cells (Fig 3A, lower part). Although MosMos transcript should not fold into dsRNA, it is possible that cells expressing high PKR levels were sensitized to dsRNA such that some cryptic transcription from the plasmid backbone could cause the effect (Nejepinska et al, 2012c).

Effects of ectopic expression of dsRBPs in ESCs expressing normal Dicer (Dicer$^S$) were similar to 3T3 cells—none of the dsRPBs had an apparent stimulatory effect on RNAi (Fig 3B). However, Dicer$^O$-expressing cells showed that ectopically expressed TARBP2 and PACT suppressed RNAi comparably with or even more than PKR (Fig 3C). This was counterintuitive because a homolog of TARBP2 and PACT is involved in RNAi in *Drosophila* (Cenik et al, 2011; Nishida et al, 2013; Fukunaga & Zamore, 2014) and TARBP2 was shown to stimulate siRNA production in vitro (Chakravarthy et al, 2010). TARBP2 and PACT could exert negative effects on RNAi in three ways: (1) by competing with Dicer's own dsRBD in recognition of dsRNA substrate, (2) by a direct inhibition of Dicer, or (3) by squelching dsRNA recognition and Dicer cleavage such that Dicer would have a reduced probability to bind its dsRBP partner bound to dsRNA.

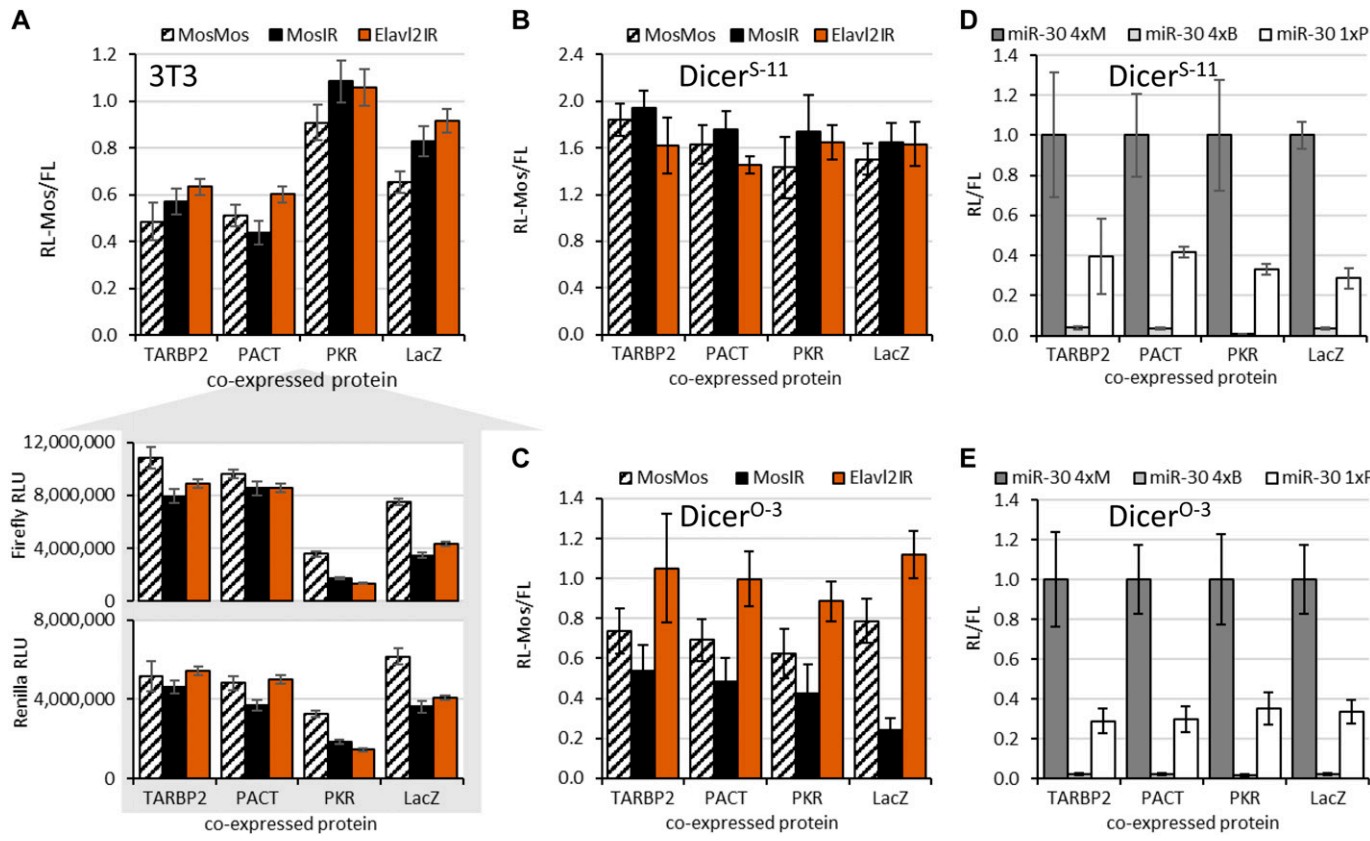

**Figure 3. Effects of expression of dsRBPs on RNAi.**
**(A)** 3T3 cells transfected with the MosIR and controls. **(B)** Dicer[S] ESCs transfected with the MosIR and controls. **(C)** Dicer[O] ESCs transfected with the MosIR and controls. **(D)** miRNA-mediated repression in Dicer[S] with increased expression of dsRPBs. **(E)** miRNA-mediated repression in Dicer[O] ESCs with increased expression of dsRPBs. Error bars in all graphs = SD. **(A–C)** Data represent at least three independent experiments performed in triplicates. **(D, E)** Data come from one transfection experiment performed in triplicates.

A simple masking appears counterintuitive as TARBP2 and PACT are Dicer-binding partners, so they would be expected to recruit Dicer to dsRNA rather than prevent its processing by Dicer. An insight into the phenomenon could be provided by miRNA reporters, which could reveal if TARBP2 or PACT overexpression interferes with miRNA-mediated repression. We did not observe any effect of TARBP2 or PACT on miRNA-mediated repression in neither Dicer[S]-expressing ESCs (Fig 3D) nor in Dicer[O]-expressing cells (Fig 3E). These data suggests that RNAi inhibition observed in Fig 3C is unlikely to involve direct Dicer inhibition, and it neither affects non-processive cleavage of miRNA precursors nor the subsequent small RNA loading. We speculate that apart from the masking effect (reduced substrate recognition), the inhibition may concern the initial internal dsRNA cleavage by Dicer or overexpressed TARBP2 affects Dicer processivity.

### Effects of dsRNA sequence and structure on RNAi

Finally, we examined the structural context of expressed dsRNA as it plays a role in efficiency of siRNA production. As mentioned earlier, Dicer preferentially cleaves dsRNA at termini and prefers single-stranded two nucleotide 3′ overhangs or blunt ends, whereas longer single-stranded RNA overhangs have an inhibitory effect and siRNA biogenesis requires endonucleolytic cleavage that occurs with lower efficiency than cleavage at dsRNA ends (Provost et al, 2002; Vermeulen et al, 2005; Zhang et al, 2002). dsRNA can be expressed in three ways, which differ in probability of dsRNA formation: (1) transcription of an inverted repeat yielding RNA hairpin, (2) convergent transcription of one sequence, and (3) separate transcription of sense and antisense strands. Furthermore, pol II and pol III transcription will yield different RNA termini, which could influence RNA localization and routing into different pathways. Therefore, we prepared a set of constructs for expression of different types of long dsRNA in addition to MosIR (CAG-EGFP-MosIR) and Lin28IR (CAG-EGFP-Lin28IR) plasmids (Fig 4). CMV-MosIR and CMV-Lin28IR are simple pol II-expressed hairpin transcripts without coding capacity; a similar expression plasmid was successfully used to induce RNAi in human HeLa cells, embryonic carcinoma cells, melanoma cells, and primary fibroblasts (Paddison et al, 2002; Diallo et al, 2003). Separate expression of sense and antisense transcripts was used to induce RNAi in NIH 3T3 and HEK 293 cell lines (Wang et al, 2003). Polyadenylated pol II transcripts would carry a 5′ cap and a single-stranded polyA 3′ overhang. To make dsRNAs with blunt (or nearly blunt) ends, we prepared also a set of pol III constructs, which included a hairpin, a convergent transcription system, and a separate sense and antisense expression. A pol

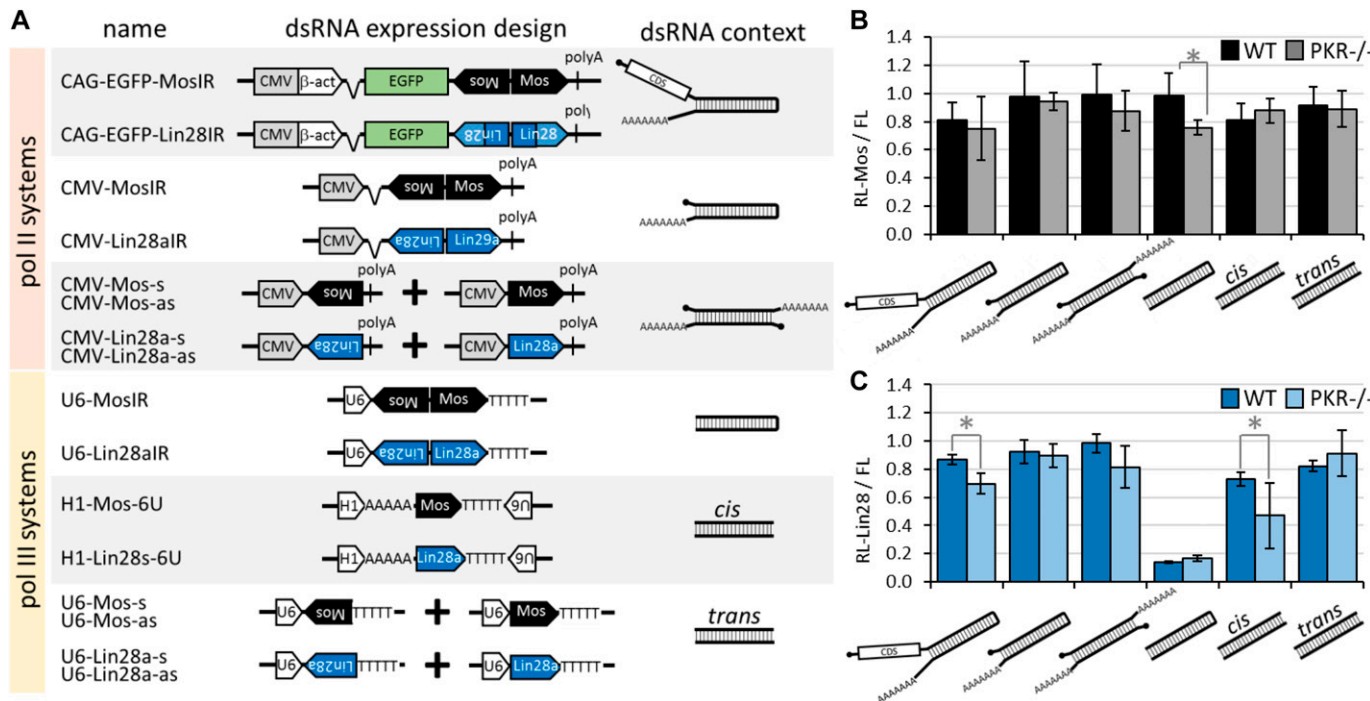

**Figure 4. Different dsRNA expression strategies have different RNAi potential.**
**(A)** Schematic depiction of relevant expressed portions of plasmids used for dsRNA expression and types of dsRNAs produced. **(B)** Impact of *Mos* dsRNA on RL-Mos in normal and *Pkr*$^{-/-}$ 3T3 cells. 3T3 cells were transfected with FL, RL-Mos reporters, and a *Mos* dsRNA-expressing plasmid or its *Lin28* counterpart. Relative *Renilla* luciferase activity (RL-Mos/FL) in *Mos* dsRNA-expressing cells is shown relative to that in *Lin28* dsRNA-expressing cells for each dsRNA type and genotype. The experiment was performed twice in triplicates. Error bars = SD. **(C)** Impact of *Lin28* dsRNA on RL-Lin28 in normal and PKR$^{-/-}$ 3T3 cells. 3T3 cells were transfected with FL, RL-Lin28, and *Lin28* dsRNA-expressing plasmid or its *Mos* counterpart. Relative *Renilla* luciferase activity (RL-Lin28/FL) in *Lin28* dsRNA-expressing cells is shown relative to that in *Mos* dsRNA-expressing cells for each dsRNA type and genotype. The experiment was performed twice in triplicates. Error bars = SD. Note that loss of *Pkr* has a small, mostly statistically insignificant, positive effect on RNAi in some cases, pol III–driven *Lin28* hairpin is inducing strong RNAi effect regardless of the presence or absence of PKR.

III–driven hairpin and a separate sense and antisense RNA expression were previously used in MCF-7 mammalian cells with little if any induction of RNAi but strong effects on the IFN response (Gantier et al, 2007).

Experimental design for testing different long dsRNAs in transiently transfected 3T3 cells was as described above (Fig 1). In addition, in an attempt to increase efficiency of RNAi, we also examined effects in *Pkr*$^{-/-}$ 3T3 cells, which were produced using CRISPR-Cas9 (Fig S4). Using *Mos* sequences, we only observed slight RNAi effects of U6-MosIR in *Pkr*$^{-/-}$ background (Fig 4C). Whether elimination of any of the other dsRNA sensors (such as *Mda5*, *Rig-I*, or *Lgp2*) in the IFN pathways would have a positive effect remains to be tested. Remarkably, U6-Lin28aIR, which expressed the same type of dsRNA hairpin, had a strong RNAi effect even in normal 3T3 (Fig 4C). Apart from U6-MosIR, U6-Lin28aIR, and U6-driven convergent transcription of *Lin28a* fragment, none of the other dsRNA substrates showed stronger induction of RNAi than CAG-EGFP-MosIR and CAG-EGFP-Lin28IR (Fig 4B and C).

To understand the basis of RNAi induction, we analyzed siRNAs produced in 3T3 cells transiently transfected with U6-MosIR, U6-Lin28aIR, CAG-EGFP-MosIR, or CAG-EGFP-Lin28IR (Figs 5 and S5). Analysis of siRNAs originating from hairpin transcripts from these plasmids showed that U6 plasmids generate 3–4 times more siRNAs than CAG-EGFP plasmids, whose dsRNA sequence is longer (Fig 5A).

Furthermore, U6 plasmids, which generate RNA hairpins with minimal single-stranded overhangs, if any, yielded a completely different pattern of siRNAs targeting RL reporters than CAG-driven hairpins, which contain long single-stranded overhangs (Fig 5B and C). U6-driven hairpins were apparently processed by Dicer from the end of the stem, which is consistent with Dicer activity in vitro (Zhang et al, 2002). The siRNA produced from U6-driven hairpins also show low processivity of the full-length mouse Dicer, where most of the reporter-targeting siRNAs come from the first substrate cleavage at the end of the stem (Fig 5B). In contrast, siRNAs produced from CAG-driven hairpins are distributed along the hairpin, suggesting that their biogenesis required at least two cleavage events—an endonucleolytic inside the stem, which produces optimal termini for a second, siRNA-producing cleavage. These two modes of siRNA production manifest distinct patterns in the phasing analysis of siRNAs produced from the two types of hairpins (Fig 5D).

Endogenous RNAi triggered by endogenous dsRNA produced in the nucleus would typically involve dsRNA molecules with long single-strand overhangs. Efficient RNAi would, thus, either require a high Dicer activity, such as the one that evolved in mouse oocytes, or some RNase activity, which would remove the single-strand overhangs, allowing for efficient Dicer cleavage from a terminus. One candidate for such an RNase is Drosha, an RNase III-producing

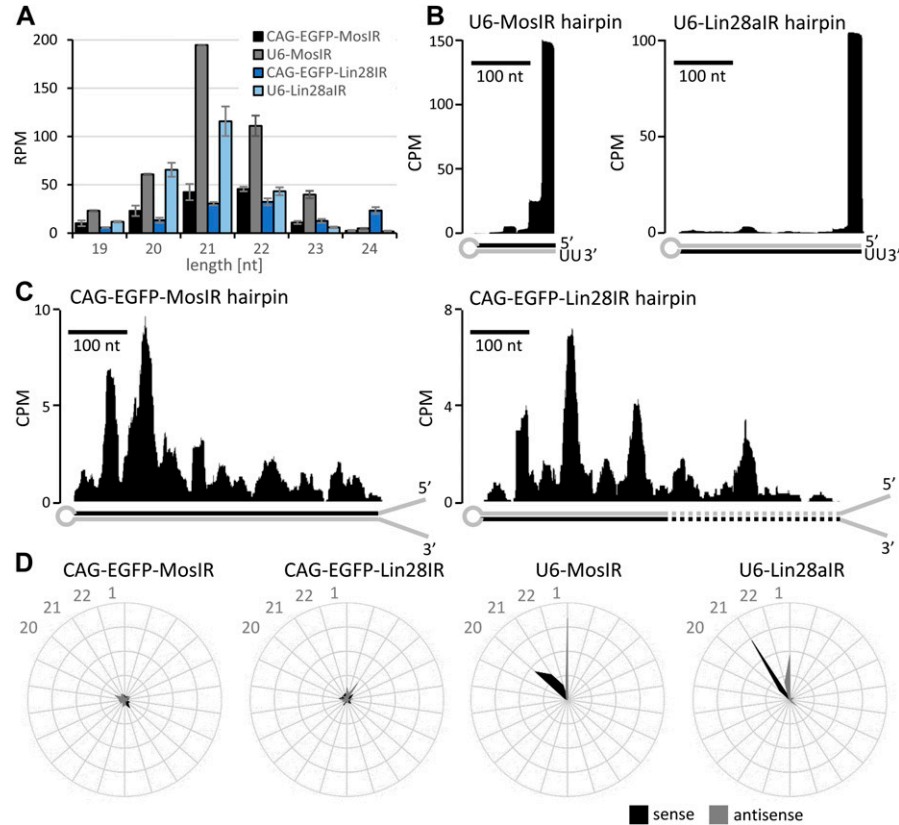

**Figure 5. siRNA production analysis from different dsRNA templates.**
**(A)** RNA-seq analysis of 3T3 cells transfected with indicated plasmids expressing different dsRNAs. Abundance of 19–24-nt small RNAs perfectly mapping to hairpins expressed from given plasmids is shown. Small RNA levels are expressed as RPM of mapped reads for each RNA-seq library perfectly mapping to the genome and transfected plasmid sequences. Libraries were prepared in duplicates from separately transfected wells. Error bars = range of data. **(B)** Graphs depicting densities of 21–23-nt reads derived from CAG-EGFP-MosIR and CAG-EGFP-Lin28IR, which are antisense to corresponding RL-Mos and RL-Lin28 reporters, respectively. Hairpin strands, which are giving rise to these siRNAs are indicated with black color. The y-scale shows read density in counts per million (CPM) normalized to the size of each library comprising all reads perfectly mapping to the genome and a transfected plasmid. Dashed lines depict *Lin28b* hairpin segment. **(C)** Distribution of reporter-targeting siRNAs produced from U6-MosIR and U6-Lin28aIR hairpins, graphs are organized as in the panel (B). **(D)** Phasing analysis of siRNAs derived from expressed hairpins.

Dicer substrate in the miRNA pathway. In this case, resulting siRNAs could be considered non-canonical miRNAs and, consequently, RNAi an extension of the miRNA pathway.

Importantly, when a blunt-end dsRNA triggers RNAi, silencing strongly depends on the first end–derived siRNA, which needs to be efficiently loaded onto AGO2 and effectively interact with the target sequence whose secondary structure can interfere with targeting (Ameres et al, 2007). This is the Achilles heel of mammalian RNAi. Evolution of endogenous regulations could be driven by positive selection of substrates mimicking miRNA features. However, for antiviral RNAi, the logic is inversed. Viruses would evolve viral variants resistant to Dicer processing of their termini into functional siRNAs. Ensuing arms race between viruses and defense mechanism would accelerate evolution of Dicer. However, vertebrate Dicer and AGO2 are well conserved (Murphy et al, 2008), arguing for their conserved role in the miRNA pathway. The known Dicer modification identified in mouse oocytes (Flemr et al, 2013), where the miRNA pathway is unimportant (Suh et al, 2010), appears to be an isolated event that occurred in the common ancestor of mice and hamsters and evolved into RNAi suppressing retrotransposons and regulating gene expression; there is no evidence it would play an antiviral role.

Taken together, we developed and examined a complex plasmid-based system for dsRNA expression in a mammalian cell and monitoring sequence-specific and sequence-independent dsRNA effects. Our plasmid collection with accompanying data offers a framework for future studies on endogenous RNAi. Our results accentuate key constraints, which influence canonical RNAi in mammalian cells and which would shape evolution of mammalian RNAi. Functional mammalian RNAi requires at least three conditions, which may be rarely present together beyond in mouse oocytes: circumventing IFN pathway sensors, sufficient dsRNA substrate, and Dicer activity necessary to yield enough effective siRNAs.

# Materials and Methods

### Plasmids

Schematic structures of the relevant parts of plasmid constructs used in the project are shown in Figs 1A and 4A. Three dsRNA-expressing plasmids (MosIR, Lin28IR, and Elavl2IR), which efficiently induced RNAi in oocytes of transgenic mice (Stein et al, 2003b; Chalupnikova et al, 2014; Flemr et al, 2014) were modified by replacing the oocyte-specific ZP3 promoter with a strong ubiquitous CAG promoter as described previously (Nejepinska et al, 2014). pGL4-SV40 (Promega; for simplicity referred to as FL) and the parental plasmid for targeted *Renilla* reporter phRL-SV40 (Promega; for simplicity referred to as RL) are commercially available. All plasmids were verified by sequencing. Non-commercial plasmids depicted in Figs 1A and 4A and plasmids expressing HA-tagged TARBP2, PACT, PKR, and LacZ

are available from Addgene with details about their construction and sequence.

## Cell culture and transfection

Mouse 3T3 cells were maintained in DMEM (Sigma-Aldrich) supplemented with 10% fetal calf serum (Sigma-Aldrich), penicillin (100 U/ml; Invitrogen), and streptomycin (100 $\mu$g/ml; Invitrogen) at 37°C and 5% $CO_2$ atmosphere. Mouse ESCs were cultured in 2i-LIF media: DMEM supplemented with 15% fetal calf serum, 1× L-glutamine (Invitrogen), 1× nonessential amino acids (Invitrogen), 50 $\mu$M $\beta$-mercaptoethanol (Gibco), 1,000 U/ml LIF (Millipore), 1 $\mu$M PD0325901, 3 $\mu$M CHIR99021, penicillin (100 U/ml), and streptomycin (100 $\mu$g/ml). For transfection, the cells were plated on a 24-well plate, grown to 50% density, and transfected using the TurboFect in vitro Transfection Reagent or Lipofectamine 3000 (Thermo Fisher Scientific) according to the manufacturer's protocol. The cells were co-transfected with 50 ng per well of each FL and RL reporter plasmids and 250 ng per well of a dsRNA-expressing plasmid and, eventually, 250 ng per well of a plasmid expressing a tested factor. The total amount of transfected DNA was kept constant (600 ng/well) using pBluescript plasmid. The cells were collected for analysis 48 h post-transfection.

## Luciferase assay

Dual luciferase activity was measured according to Hampf & Gossen (2006) with some modifications. Briefly, the cells were washed with PBS and lysed in PPTB lysis buffer (0.2% vol/vol Triton X-100 in 100 mM potassium phosphate buffer, pH 7.8). 3–5-$\mu$l aliquots were used for measurement in 96-well plates using Modulus Microplate Multimode Reader (Turner Biosystems). First, firefly luciferase activity was measured by adding 50 $\mu$l substrate (20 mM tricine, 1.07 mM $(MgCO_3)_4 \cdot Mg(OH)_2 \cdot 5\ H_2O$, 2.67 mM $MgSO_4$, 0.1 mM EDTA, 33.3 mM DTT, 0.27 mM Coenzyme A, 0.53 mM ATP, and 0.47 mM D-Luciferin, pH 7.8) and a signal was integrated for 10 s after a 2-s delay. The signal was quenched by adding 50 $\mu$l Renilla substrate (25 mM $Na_4PP_i$, 10 mM Na-acetate, 15 mM EDTA, 500 mM $Na_2SO_4$, 500 mM NaCl, 1.3 mM $NaN_3$, and 4 $\mu$M coelenterazine, pH 5.0) and *Renilla* luciferase activity was measured for 10 s after a 2-s delay.

## Western blotting

3T3 cells were grown in six-well plates. Before collection, the cells were washed with PBS and lysed in lysis buffer (20 mM Hepes [pH 7.8], 100 mM NaCl, 1 mM EDTA [pH 8.0], 0.5% IGEPAL-25%, 1 mM fresh DTT, 0.5 mM PMSF, 1 mM NaF, 0.2 mM $Na_3VO_4$, supplemented with 2× protease inhibitor cocktail set [Millipore], 2× phosphatase inhibitor cocktail set [Millipore], and RiboLock RNase inhibitor [Thermo Fisher Scientific]). Proteins were separated on 10% polyacrylamide gel and transferred to a PVFD membrane (Millipore). Anti-HA (11867431001, clone 3F10, rat, 1:2,500; Roche), anti-PKR (#ab32052, 1:5,000 dilution; Abcam), and anti-tubulin (#T6074, 1:5,000; Sigma-Aldrich) primary antibodies and anti-rat HRP (1:50,000) secondary antibody were used for signal detection with

SuperSignal West Femto or Pico Chemiluminescent Substrate (Pierce).

## Pkr knock-out in 3T3 cells

For PKR knock-out cells, exons 2–5 (~7.5-kb region coding for dsRNA-binding domains) were deleted using the CRISPR approach. sgRNAs targeting intron 1 (5′-<u>CCT</u>TCTTTAACACTTGGCTTC & 5′-<u>CCT</u>GTGGTG-GGTTGGAAACAC) and intron 5 (5′-GTGGAGTTGGTGGCCACG<u>GGG</u> & 5′-<u>CCT</u>GTGTACCAACAATGATCC) were co-transfected with Cas9-expressing and puromycin selection plasmids. After 48 h, the cells were selected with puromycin (f.c. = 3 $\mu$g/ml) for 2 d. and individual clones were isolated and screened for the presence of deletion using PCR (forward primer: 5′-GCCTTGTTTTGACCATAAATGCCG and reverse primer: 5′-GTG-ACAACGCTAGAGGATGTTCCG). Expression of PKR lacking dsRNA-binding domains was confirmed by qPCR and homozygote clones were used for further experiments.

## RNA-seq

The cells were plated on six-well plates and grown to 50% density. The cells were transfected with 2 $\mu$g/well of U6-MosIR, U6-Lin28aIR, CAG-EGFP-MosIR, or CAG-EGFP-Lin28IR plasmids, cultured for 48 h, washed with PBS, and total RNA was isolated using RNAzol (MRC) according to the manufacturer's protocol. RNA quality was verified by Agilent 2100 Bioanalyzer. The library construction and high-throughput sequencing of the RNA transcriptome were performed either from small RNA (<200 nt) fraction using SOLiD (version 4.0) sequencing platform (Seqomics) or libraries were constructed from total RNA using NEXTflex Small RNA-Seq Kit v3 (Bioo Scientific) according to the manufacturer's protocol and sequenced on the Illumina HiSeq2000 platform at the Genomics Core Facility at EMBL. High-throughput sequencing data were deposited in the GEO database (GSE41207 for SOLiD data and GSE126324 for Illumina data).

## Bioinformatic analyses

Bioinformatic analysis of SOLiD data was performed as described previously (Nejepinska et al, 2012b; Flemr et al, 2013). Briefly, SOLiD raw .csfasta and .qual files were quality filtered and trimmed using cutadapt 1.16 (Martin, 2011): *cutadapt -e 0.1 -m 15 -c -z -a 'CGCCT-TGGCCGTACAGCAG' -o ${FILE}.trim.fastq.gz $FILE.csfasta.gz $FILE.qual.gz*.

Trimmed reads were mapped in colorspace onto indexed genome using SHRiMP 2.2.3 (David et al, 2011): *gmapper-cs ${FILE}. trim. fastq.gz −threads 10 -L $REF_GENOME_INDEX -o 99999 -E −local −strata > ${FILE}.sam*.

Illumina raw .fastq files were trimmed in two rounds using bbduk 37.95 (https://jgi.doe.gov/data-and-tools/bbtools/). First, adapter was trimmed from 3′ end or reads: *bbduk.sh in=${FILE}.fastq.gz out=${FILE}.atrim.fastq.gz literal=TGGAATTCTCGGGTGCCAAGG ktrim=r k=19 rcomp=t mink=10 hdist=1 minoverlap=8*.

Next, four bases were trimmed from both 5′ and 3′ ends of adapter-trimmed reads: *bbduk.sh in=${FILE}.atrim.fastq.gz out= ${FILE}.trim.fastq.gz forcetrimright2=4 forcetrimleft=4 minlength=15*.

Trimmed reads were mapped onto indexed genome using STAR 2.5.3a (Dobin et al, 2013): *STAR –readFilesIn ${FILE}.trim.fastq.gz –genomeDir $ REF_GENOME_INDEX –runThreadN 10 –genomeLoad LoadAndRemove –limitBAMsortRAM 20000000000 –readFilesCommand unpigz –c –outFileNamePrefix ${FILE}. –outSAMtype BAM SortedByCoordinate –outReadsUnmapped Fastx –outFilterMismatchNmax 2 –outFilterMismatchNoverLmax 1 –outFilterMismatchNoverReadLmax 1 –outFilterMatchNmin 16 –outFilterMatchNminOverLread 0 –outFilterScoreMinOverLread 0 –outFilterMultimapNmax 99999 –outFilterMultimapScoreRange 0 –alignIntronMax 1 –alignSJDBoverhangMin 999999999999.*

Both SOLiD and Illumina-trimmed read files were mapped onto mouse genome version mm10/GRCm38 with plasmid sequences (available in the supplemental file as an annotated GenBank format) added to the genome before indexing.

External genomic annotations sets were used for the analysis. miRNA coordinates were downloaded from the miRBase, v22 (Kozomara & Griffiths-Jones, 2014). Exon coordinates were downloaded from Ensembl database, release 91 (Aken et al, 2017). Coordinates of repeats were downloaded as RepeatMasker (Smit et al, 2013–2015) track from UCSC genome browser (Kuhn et al, 2013). The downstream analysis was performed in the R software environment (https://www.R-project.org). Unless noted otherwise, all downstream analyses were performed with 21–23-nt long reads perfectly matching the genome sequence.

Small RNA read clusters (Fig 2C and D) were identified following the algorithm used in a previous study (Flemr et al, 2013) with few changes. In short:

1) Reads were weighted to fractional counts of 1/n where n represents the number of loci to which read maps.
2) Reads were then collapsed into a unified set of regions and their fractional counts were summed.
3) Clusters with less than three RPM were discarded.
4) Clusters within 50 bp distance of each other were joined.

Only clusters appearing in all replicates of the same genotype (intersect) were considered in the final set. Union of coordinates of overlapping clusters were used to merge the clusters between the samples. The clusters were then annotated, and if a cluster overlapped more than one functional category, the following classification hierarchy was used: miRNA > transposable elements > mRNA (protein coding genes) > misc. RNA (other RNA annotated in ENSEMBL or RepeatMasker) > other (all remaining annotated or not annotated regions).

To visualize phasing of siRNAs derived from expressed hairpins on radar plots (Fig 5D), start coordinates of all 21–23-nt reads were first scaled in regard to start of hairpin on plasmid (so that first nucleotide of hairpin defines register 1). Proportion of reads in each of 22 registers was calculated as a modulo-22 of scaled start coordinates divided by total number of reads belonging to all hairpin registers. This phasing analysis was adapted from (Maillard et al, 2013).

## Supplementary Information

# Acknowledgements

We thank Vedran Franke for help with data analysis, Vladimir Benes and EMBL sequencing facility for help with RNA-seq experiments, and Kristian Vlahovicek for providing hardware support for bioinformatics analysis. This work was funded from the European Research Council under the European Union's Horizon 2020 research and innovation programme (grant agreement No 647403, D-FENS). Additional support was provided by the Czech Science Foundation grant P305/12/G034 and by the Ministry of Education, Youth, and Sports project NPU1 LO1419. Additional computational resources for PS lab were provided by the CESNET LM2015042 and the CERIT Scientific Cloud LM2015085 under the programme "Projects of Large Research, Development, and Innovations Infrastructures."

## Author Contributions

T Demeter: data curation, formal analysis, validation, investigation, visualization, and methodology.

M Vaskovicova: data curation, formal analysis, validation, investigation, visualization, and methodology.

R Malik: conceptualization, data curation, formal analysis, supervision, validation, investigation, visualization, methodology, project administration, and writing—original draft, review, and editing.

F Horvat: data curation, software, formal analysis, validation, visualization, and methodology.

J Pasulka: data curation, software, formal analysis, validation, visualization, and methodology.

E Svobodova: resources, data curation, formal analysis, validation, and investigation.

M Flemr: resources, data curation, formal analysis, and investigation.

P Svoboda: conceptualization, data curation, supervision, funding acquisition, validation, visualization, project administration, and writing—original draft, review, and editing.

## Conflict of Interest Statement

The authors declare that they have no conflict of interest.

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
