## [Reviewer comments · Life Science Alliance]

Main constraints for RNAi induced by expressed long dsRNA in mouse cells

Tomas Demeter, Michaela Vaskovicova, Radek Malik, Filip Horvat, Josef Pasulka, Eliska Svobodova, Matyas Flemr, and Petr Svoboda

DOI: <https://doi.org/10.26508/lsa.201800289>

Corresponding author(s): Petr Svoboda, Institute of Molecular Genetics of the Czech Academy of Sciences

Review Timeline:

Submission Date:	2018-12-27
Editorial Decision:	2019-01-07
Revision Received:	2019-02-03
Editorial Decision:	2019-02-05
Revision Received:	2019-02-09
Accepted:	2019-02-11

Scientific Editor: Andrea Leibfried

Transaction Report:

Please note that the manuscript was previously reviewed at another journal and the reports were taken into account in inviting a revision for publication at *Life Science Alliance* prior to submission to *Life Science Alliance*.

Referee #1 Review

Report for Author:

In general I thought this an interesting and technically proficient manuscript that provides a careful examination of the factors that limit siRNA production in mammalian somatic cells. Though the work is perhaps somewhat predictable based on work from the tenOever and Cullen labs, the issue of viral siRNAs and whether they really exist has remained contentious, so I do favor publication. My main concern relates to the issue of why 3T3 cells transfected with MosIR and the Dicer0 construct appear to generate quite high levels of siRNAs (Fig. 2A) yet do not give much of an inhibitory effect (Fig. 2G). One possible explanation is that these siRNAs are not effectively loaded into RISC, as also reported previously for some ineffective miRNAs (Flores et al, NAR42, p4629, 2014). To address this concern, the authors need to perform small RNA-seq on RISC-associated small RNAs from these cells by pan-Ago or Ago2 IP, as previously described by several groups. If these data are provided, I would then favor publication.

Referee #2 Review

Report for Author:

In this work, Demeter et al. investigate the modalities of RNAi engagement by long dsRNA in mouse cells. This is part of a 18 year old debate as to whether or not long dsRNA expressed by somatic mammalian cells can be processed by Dicer into siRNAs to efficiently engage the RNAi pathway.

The Svoboda laboratory has already reported the existence of a unique truncation of Dicer expressed in mouse Oocytes (Dicer-o) which can process long dsRNAs into siRNAs, while full length Dicer cannot. In addition this laboratory has already shown that mice expressing the MOS inverted cassette was poorly processed into siRNA in vivo. Although very thorough, it is hard to see what the current study really adds to the topic, and the study thus appears incremental.

On one hand, the authors show detail profiling of small RNAs after expression of their constructs in NIH3T3 MEFs, with the additional context of Dicer-o/s TRBP/PACT - which is new. But none of the findings actually really add to what they have previously published.

In addition, there is a key conceptual issue in comparing overexpressed dsRNA expression (from the nucleus) from a transfected vector (which would likely activate innate immune response through cGAS - expressed in NIH3T3 cells) with what happens during viral infection where the dsRNAs can be generated directly in the cytoplasm - which therefore precludes any nuclear retention of dsRNA species, potentially at play with overexpression. If the study had incorporated transfection of synthetic dsRNAs in cells devoid of RIG-I/MDA5/PKR and looked at the processing of these RNAs into siRNAs, it would probably have been more interesting as the substrate of Dicer would have been known (but here the

assumption is made on the nature of the dsRNA being produced from the DNA constructs).

Also, it is now clear that select endogenous dsRNAs can be generated during mouse fetal liver development, and that these are strong ligands of MDA5, but editing by the dsRNA protein ADAR1 helps prevent MDA5 engagement. Since ADAR1 is essential to limit accumulation of these dsRNA, it seems obvious that Dicer alone is not able to significantly process such dsRNAs (PMID:26275108).

The authors also failed to mention a recent study strongly supporting the antiviral role of the RNAi pathway against Influenza A virus - where Ago2 mutant mice exhibited increased viral production (PMID:27918527).

Altogether, what the paper was actually attempting to demonstrate was not clear and the line of thought was also hard to follow. The last paragraph exemplifies this point - the summary of what the manuscript shows is very confusing.

Referee #3 Review

Report for Author:

In this study, Demeter and co-authors investigate the efficiency of RNAi in mouse cells. The experimental approach is straightforward and involves the analysis of RNAi induction in mouse fibroblasts and ESCs using a set of plasmids expressing different dsRNAs that target luciferase reporters.

It had been previously shown that a way to increase the efficiency of RNAi in mammalian cells implied truncation of Dicer at its N-terminus. Interestingly, an N-terminally truncated Dicer variant is present in mouse oocytes and consequently RNAi is highly active and functionally important.

The major finding of this study is that RNAi is restricted at multiple levels in mouse cells and that RNAi is not operating in mouse 3T3 cells. The authors observed that low endogenous Dicer activity only yields limited amounts of siRNA from dsRNA, unless the substrate contains a terminus resembling that of a miRNA precursors.

Demeter and colleagues made the unexpected observation that an increased expression of dsRNA-binding proteins TARBP2 or PACT, which act as Dicer co-factors, reduces RNAi but does not influence miRNA function. Mechanistically, it remains unclear how the increased expression of PACT and/or TARBP2 lead to RNAi inhibition.

The experiments have been very well designed and most of the results are clear, although some show marginal effects. Overall, there are some interesting observations here; however, the degree of novelty or mechanistic insights into the mechanism of RNAi in mouse cells seems perhaps borderline for publication in this journal.

January 7, 2019

Re: Life Science Alliance manuscript #LSA-2018-00289-T

Petr Svoboda
Institute of Molecular Genetics of the Czech Academy of Sciences
Laboratory of Epigenetic Regulations
Videnska 1083
Prague 14220
Czech Republic

Dear Dr. Svoboda,

Thank you for transferring your manuscript entitled "Main constraints for RNAi induced by expressed long dsRNA in mouse cells" to Life Science Alliance. The manuscript was assessed by expert reviewers at another journal before, and the editors transferred those reports to us with your permission.

The reviewers thought that your data are robust, but they would have expected further reaching insight and a broader conceptual advance. This is not a concern for publication here, and I would thus like to invite you to submit a slightly revised version. We would expect a point-by-point response to the concerns raised. The limitations noted by the reviewers should get discussed in the manuscript and the narrative should get altered to allow an easier read. While it would be also good to test RISC loading as suggested by reviewer #1, I understand that this experiment is difficult to perform. A thorough discussion and addressing the discrepancy this reviewer noted in the text is therefore sufficient for publication here.

The typical timeframe for revisions is three months.

Thank you for this interesting contribution to Life Science Alliance. We are looking forward to receiving your revised manuscript.

Sincerely,

Andrea Leibfried, PhD
Executive Editor
Life Science Alliance
Meyerhofstr. 1

69117 Heidelberg, Germany
t +49 6221 8891 502
e a.leibfried@life-science-alliance.org
www.life-science-alliance.org

- A letter addressing the reviewers' comments point by point.
- An editable version of the final text (.DOC or .DOCX) is needed for copyediting (no PDFs).
- High-resolution figure, supplementary figure and video files uploaded as individual files: See our detailed guidelines for preparing your production-ready images, <http://life-science-alliance.org/authorguide>
- Summary blurb (enter in submission system): A short text summarizing in a single sentence the study (max. 200 characters including spaces). This text is used in conjunction with the titles of papers, hence should be informative and complementary to the title and running title. It should describe the context and significance of the findings for a general readership; it should be written in the present tense and refer to the work in the third person. Author names should not be mentioned.

B. MANUSCRIPT ORGANIZATION AND FORMATTING:

Full guidelines are available on our Instructions for Authors page, <http://life-science-alliance.org/authorguide>

Referee #1:

My main concern relates to the issue of why 3T3 cells transfected with *MosIR* and the *Dicer*⁰ construct appear to generate quite high levels of siRNAs (Fig. 2A) yet do not give much of an inhibitory effect (Fig. 2G). One possible explanation is that these siRNAs are not effectively loaded into RISC, as also reported previously for some ineffective miRNAs (Flores et al, NAR42, p4629, 2014). To address this concern, the authors need to perform small RNA-seq on RISC-associated small RNAs from these cells by pan-Ago or Ago2 IP, as previously described by several groups.

Profiles of 21-23 nt small RNAs mapped onto the *MosIR* sequence were very similar regardless of the cell type, *Dicer* isoform, and RNAi efficiency (Fig. S2B and PMID: 24209619). Inefficient RISC loading should be accompanied by accumulation of siRNA duplexes but these data show that passenger strands of siRNAs do not seem to accumulate in any of the sequenced samples.

While abundance of *Mos* siRNAs in RNA-seq data was higher than one would expect from moderate suppression of the targeted reporter, this could be a consequence of several features of the experimental setup. First, as we discuss in the manuscript, only a fraction of siRNAs depicted in Fig. 2A would efficiently target *Mos* reporter – statistically, only ½ of loaded siRNAs would be antisense to the reporter RNA and only a fraction of those would be efficient siRNAs (i.e not every AGO2-loaded 21-23 nt antisense siRNA would efficiently suppress the target). Second, some bias may come from sample heterogeneity in transient co-transfection of 3T3 cells where *Dicer*⁰ was also expressed from a co-transfected plasmid. Third, we cannot rule out that poorly folded *MosIR* transcripts could serve as a decoy and reduce efficiency of reporter targeting (this we did not mention and we add to the discussion now). At the same time, the *Mos*-based RNAi system is excellent for studying constraints for endogenous RNAi because it requires optimized RNAi activity while Lin28-based system appears to be more sensitive for detecting RNAi.

In any case, our results show that constructs, which can induce endogenous RNAi in mouse oocytes, do not efficiently induce RNAi in an ectopic expression system because of poor siRNA production by endogenous *Dicer*. As performing the proposed AGO-IP experiment did not seem to be critical because of the similarity of small RNA profiles, we just revised discussion of this point for revision for LSA.

Referee #2:

In this work, Demeter et al. investigate the modalities of RNAi engagement by long dsRNA in mouse cells. This is part of a 18 year old debate as to whether or not long dsRNA expressed by somatic mammalian cells can be processed by *Dicer* into siRNAs to efficiently engage the RNAi pathway.

One of our motivations for publishing this work and its integral part were to share a well-characterized set of plasmids expressing dsRNA and sensing sequence-specific & sequence-independent effects. This material can provide a kind of benchmark, at least for experiments with expressed dsRNA. All three long dsRNA hairpins were proven to induce RNAi *in vivo* in mouse

oocytes. Combining them with reporters, we provide basic characterization of RNAi induction, siRNA production, which includes negative data and shows that in a unique case, RNAi can be induced by long dsRNA expression in a somatic cell type without enhanced Dicer activity.

The Svoboda laboratory has already reported the existence of a unique truncation of Dicer expressed in mouse Oocytes (Dicer-o) which can process long dsRNAs into siRNAs, while full length Dicer cannot. In addition this laboratory has already shown that mice expressing the MOS inverted cassette was poorly processed into siRNA *in vivo*. Although very thorough, it is hard to see what the current study really adds to the topic, and the study thus appears incremental.

We accept that the study can be seen as incremental but we believe it is important to share these data nonetheless. Our contribution provides a systematic background, which was somewhat missing among the publications reporting various observations of RNAi induced with long dsRNA where negative results were probably not discussed as often as they were observed. We accumulated a volume of negative data implying that reports of efficient mammalian canonical RNAi represent specific conditions rather than depicting widespread RNAi presence. This would also concern reports from mouse embryonic stem cells. In this paper, we wish to share these negative data and build them into the picture of mammalian RNAi.

On one hand, the authors show detail profiling of small RNAs after expression of their constructs in NIH3T3 MEFs, with the additional context of Dicer-o/s TRBP/PACT - which is new. But none of the findings actually really add to what they have previously published.

In our view, the major addition to what we have previously published regarding endogenous RNAi in mice is showing:

- (1) how unlikely yet possible is RNAi induced with long dsRNA produced by pol II in the nucleus.
- (2) unexpected inhibitory effects of overexpressed Dicer binding partners TARBP2 and PACT
- (3) that blunt-end dsRNA can induce RNAi without enhancing Dicer activity or suppression of PKR but, at the same time, it is a matter of probability that the first siRNA produced from a dsRNA terminus will mediate efficient RNAi. In a sense, it is the same kind of probability of synthesizing a functional siRNA from a random 21-23 nt region of a cognate sequence. This is something what is not necessarily new but we think should be highlighted in the ongoing discussion about potential roles of mammalian endogenous RNAi.

If the study had incorporated transfection of synthetic dsRNAs in cells devoid of RIG-I/MDA5/PKR and looked at the processing of these RNAs into siRNAs, it would probably have been more interesting as the substrate of Dicer would have been known (but here the assumption is made on the nature of the dsRNA being produced from the DNA constructs).

Testing siRNA biogenesis from transfected dsRNA in cells devoid of dsRNA sensors is a valid line of research, which is, however, oriented towards the antiviral/innate immunity function of RNAi. Our experimental design aims at endogenous mammalian RNAi, which is induced by

expressed dsRNA and targets endogenous transcripts – mRNAs and retrotransposon RNAs. As this pathway is a known adaptation in mouse oocytes, we aim at understanding what are the functional limits of this pathway beyond mouse oocytes. Transfection of synthetic dsRNA represents entirely different experimental setup mimicking viral infections - it is based on dsRNA formed outside of the cell and appearing in the cytoplasm without nuclear history. Off note is that, when it comes to sequence-independent effects of the interferon response, our system seems to be primarily sensed by PKR and we almost always observe sequence independent effects depicted in Fig.1 B, which are not visible when data are normalized to the control firefly luciferase reporter. We believe that this is an important phenomenon, which should be pointed out repeatedly.

Also, it is now clear that select endogenous dsRNAs can be generated during mouse fetal liver development, and that these are strong ligands of MDA5, but editing by the dsRNA protein ADAR1 helps prevent MDA5 engagement. Since ADAR1 is essential to limit accumulation of these dsRNA, it seems obvious that Dicer alone is not able to significantly process such dsRNAs (PMID:26275108).

We studied PKR and ADAR1 impact on RNAi in embryonic stem cells. While deletion of PKR (exons 2-5) had a strong positive effect on RNAi efficiency, deletion of ADAR1 (CRISPR-mediated deletion of exons 2-6) did not. These data will be published separately, as a part of our effort on enhancing RNAi *in vivo* through genetic modifications of endogenous genes.

We showed previously in cultured human HEK239 cells (PMID: 24475301) that up to ½ siRNAs could be edited. In 3T3 cells, however, mapping only perfectly matching reads did not yield dramatically lower count of mapped reads than single or two mismatch-permitting mapping of putative *Mos* siRNAs (the difference was less than 15%). In case of editing, one would expect that a fraction of reads mapping with mismatches could be edited. Thus, it does not seem that adenosine deamination was a major player in our particular experimental set up.

To address this reviewer's comment in the revised version, introduction lists MDA5 among RIG-I-like receptors (RIG-I, MDA5, and LGP2), and we added a comment into results and discussion pointing out that we did not test the role of RIG-I-like dsRNA sensors.

The authors also failed to mention a recent study strongly supporting the antiviral role of the RNAi pathway against Influenza A virus - where Ago2 mutant mice exhibited increased viral production (PMID:27918527).

We included the recent study in the introduction. Off note is that HEK293 cells contain very high levels of Dicer (3-4x more than ESCs , PMID: 20730047) which is in agreement with the idea that Dicer level influences ability to produce siRNA and that RNAi may become functional under specific circumstances. We would like to point out that we are not arguing that RNAi can't work, we argue in the light of the volume of negative data that functional RNAi is rather an exceptional situation than a rule.

Altogether, what the paper was actually attempting to demonstrate was not clear and the line of thought was also hard to follow.

The paper was examining why canonical endogenous RNAi induced by expressed long dsRNA rarely works in mouse somatic cells, while there are publications claiming efficient and specific RNAi effects. Thus, our line of thought was to test a number of conditions in a more systematic way, which could provide a benchmark for future studies and would reveal that we also can induce RNAi, but only under specific circumstances. We revised the introduction and conclusions to make this point more apparent.

The last paragraph exemplifies this point - the summary of what the manuscript shows is very confusing.

The last paragraph argues that RNAi is an unlikely mammalian antiviral mechanism because the available Dicer activity available in somatic cells can generate siRNAs essentially only from the very end of a blunt-end of abundant dsRNA. As mentioned above, we revised the introduction and conclusions to make this point more apparent.

Referee #3:

The experiments have been very well designed and most of the results are clear, although some show marginal effects. Overall, there are some interesting observations here; however, the degree of novelty or mechanistic insights into the mechanism of RNAi in mouse cells seems perhaps borderline for publication in this journal.

We appreciate with reviewer's comment. In terms of the data quality and interpretation, there was no specific point raised, which would require revision for LSA.

February 5, 2019

RE: Life Science Alliance Manuscript #LSA-2018-00289-TR

Prof. Petr Svoboda
Institute of Molecular Genetics of the Czech Academy of Sciences
Laboratory of Epigenetic Regulations
Videnska 1083
Prague 14220
Czech Republic

Dear Dr. Svoboda,

Thank you for submitting your revised manuscript entitled "Main constraints for RNAi induced by expressed long dsRNA in mouse cells". I appreciate the introduced changes and would thus be happy to publish your paper in Life Science Alliance pending final revisions necessary to meet our formatting guidelines:

- I would like to suggest re-writing of abstract and running title, see suggestion below
- please deposit the RNA-seq data and provide a data availability statement
- please add a callout in the ms text to Fig 4B and Fig S2A

A. FINAL FILES:

-- High-resolution figure, supplementary figure and video files uploaded as individual files: See our detailed guidelines for preparing your production-ready images, <http://life-science-alliance.org/authorguide>

B. MANUSCRIPT ORGANIZATION AND FORMATTING:

Full guidelines are available on our Instructions for Authors page, <http://life-science-alliance.org/authorguide>

Sincerely,

Suggested alternative title, abstract and running title:

Main constraints for RNAi in response to expressing long dsRNA in mouse cells

Running title: Efficiency of RNAi in mammalian cells

RNA interference (RNAi) is the sequence-specific mRNA degradation guided by small RNAs (siRNAs) produced from long double-stranded RNA (dsRNA) by RNase Dicer. Proteins executing RNAi are present in mammalian cells but rather sustain the microRNA pathway. Aiming for a systematic analysis of mammalian RNAi, we report here that the main bottleneck for RNAi efficiency is the production of functional siRNAs, which integrates Dicer activity, dsRNA structure, and siRNA targeting efficiency. Unexpectedly, increased expression of Dicer co-factors TARBP2 or PACT reduces RNAi but not microRNA function. Elimination of Protein Kinase R, a key dsRNA sensor in the interferon response, had minimal positive effects on RNAi activity in fibroblasts. Without high Dicer activity, RNAi can still occur when the initial Dicer cleavage of the substrate yields an efficient siRNA. Efficient mammalian RNAi may employ substrates with some features of microRNA precursors, merging both pathways even more than previously suggested. While optimized endogenous Dicer substrates mimicking miRNA features could evolve for endogenous regulations, the same principles would make antiviral RNAi inefficient as viruses would adapt to avoid efficacy.

February 11, 2019

RE: Life Science Alliance Manuscript #LSA-2018-00289-TRR

Prof. Petr Svoboda
Institute of Molecular Genetics of the Czech Academy of Sciences
Laboratory of Epigenetic Regulations
Videnska 1083
Prague 14220
Czech Republic

Dear Dr. Svoboda,

Thank you for submitting your Research Article entitled "Main constraints for RNAi induced by expressed long dsRNA in mouse cells". It is a pleasure to let you know that your manuscript is now accepted for publication in Life Science Alliance. Congratulations on this interesting work.

DISTRIBUTION OF MATERIALS:

Again, congratulations on a very nice paper. I hope you found the review process to be constructive and are pleased with how the manuscript was handled editorially. We look forward to future exciting submissions from your lab.

Sincerely,
